# Estimating Throwing Speed in Handball Using a Wearable Device

**DOI:** 10.3390/s20174925

**Published:** 2020-08-31

**Authors:** Sebastian D. Skejø, Jesper Bencke, Merete Møller, Henrik Sørensen

**Affiliations:** 1Department of Public Health, Aarhus University, 8000 Aarhus, Denmark; sdsk@ph.au.dk; 2Human Movement Analysis Laboratory, Department of Orthopaedic Surgery, Copenhagen University Hospital Amager-Hvidovre, 2650 Hvidovre, Denmark; jesper.bencke@regionh.dk; 3Department of Sports Science and Clinical Biomechanics, University of Southern Denmark, 5230 Odense, Denmark; memoller@health.sdu.dk

**Keywords:** accelerometer, throwing velocity, inertial measurement unit, throwing load, shoulder load

## Abstract

Throwing speed is likely a key determinant of shoulder-specific load. However, it is difficult to estimate the speed of throws in handball in field-based settings with many players due to limitations in current technology. Therefore, the purpose of this study was to develop a novel method to estimate throwing speed in handball using a low-cost accelerometer-based device. Nineteen experienced handball players each performed 25 throws of varying types while we measured the acceleration of the wrist using the accelerometer and the throwing speed using 3D motion capture. Using cross-validation, we developed four prediction models using combinations of the logarithm of the peak total acceleration, sex and throwing type as the predictor and the throwing speed as the outcome. We found that all models were well-calibrated (mean calibration of all models: 0.0 m/s, calibration slope of all models: 1.00) and precise (R2 = 0.71–0.86, mean absolute error = 1.30–1.82 m/s). We conclude that the developed method provides practitioners and researchers with a feasible and cheap method to estimate throwing speed in handball from segments of wrist acceleration signals containing only a single throw.

## 1. Introduction

The prevalence and incidence of shoulder problems are very high in handball [1,2,3]. In order to understand why and how such problems occur, it is required that causal pathways between exposures and outcomes are established [4]. Changes in training load are an example of an often-used exposure that is causally linked to injury development [5]. Following this line of thought, Møller et al. [1] showed that sudden increases in the number of hours spent playing handball is the primary driver of the development of shoulder problem. However, it seems likely that when investigating shoulder-specific problems, a more direct estimation of shoulder-specific load may lead to an increased understanding of why such problems occur.

Shoulder-specific load as opposed to training load in general, is a term used here to describe the overall loads of the structures of the shoulder [4]. It seems biomechanically plausible that the throwing speed could be a key determinant of the shoulder-specific load: Higher throwing speeds require larger internal rotation moments at the shoulder [6], larger internal rotation moments at the shoulder require larger muscle forces and ultimately, larger muscle forces cause larger amounts of micro-damage at the tissue level [7]. To our knowledge, there are no studies specifically investigating the relation between throwing speed and shoulder injury risk, but two baseball studies found an association between pitching speed and elbow injury [8,9]. As such, there are indications that monitoring the speed of throws is important in order to understand and prevent shoulder injuries in handball.

Additionally, previous research has suggested that return-to-throwing rehabilitation programs should control the throwing intensity [10] and recently Lizzio et al. [11] showed that the internal rotation moment at the shoulder is better controlled when controlling throwing speed rather than perceived throwing effort. Thus, a method for estimating throwing speed could also help practitioners when rehabilitating shoulder injuries.

Unfortunately, no cheap and feasible method that can be applied in the field and in large cohorts currently exists. Existing methods for measuring the speed of throws in general include camera-based tracking methods and commercial wearable devices. However, camera-based methods are typically both expensive and requires a multi-camera setup, and to our knowledge, no existing wearable device has been validated in a handball setting. Nonetheless, wearable devices have been used to estimate ball speed in other sports [12] and given the low cost of wearable devices, such as accelerometers, it would be interesting to investigate the possibility of using accelerometers to determine speed of throws in handball as well.

It should be noted that the shoulder-specific load also depends on the number of throws as well as other complex interactions but measuring the number of throws and the speed of each throw are two separate problems. The former requires extracting the parts of a long recording of accelerations that contain a throw. The latter requires transforming the extracted parts into a single number representing the speed of the throw. In the present study, we are only concerned with solving the latter problem. Thus, the purpose of this study is to develop a novel accelerometer-based device that can estimate the speed of handball throws. We find that the suggested method is valid and well-calibrated, indicating that the method can be used to estimate throwing speed in handball.

## 2. Materials and Methods

### 2.1. Subjects

We recruited 19 experienced handball players (8 females, 11 males, age: 19 to 33 years, mass: 60.2–116.7 kg, height: 1.60–1.96 m, level of play ranging from amateur to professional) to participate in this cross-sectional study. Participants had to be at least 18 years old and not have experienced shoulder pain within the last month. The local research ethics committee waived the need for ethical approval and all participants provided informed consent prior to participating in the present study.

### 2.2. Methodology

All trials were performed in a biomechanical laboratory. Following a standardized warm-up, each participant performed a total of 25 throws using five different types of throws: (i) low-intensity standing throw without run-up, (ii) medium-intensity standing throw without run-up, (iii) maximum-intensity standing throw without run-up, (iv) maximum-intensity standing throw with run-up and (v) maximum-intensity jump throw with run-up. Each technique was used for five throws and the order of throws was randomised. For the low-, medium- and maximum-intensity standing throws without run-up, we asked the participants to emulate a short pass, a long pass, and to throw as fast as possible, respectively.

During each throw, we recorded the position of a reflective marker on both the hand and the ball using 3D motion capture (8 cameras recording at 240 Hz; ProReflex MCU1000, Qualisys AB, Gothenburg, Sweden). Simultaneously, a custom-built device containing a triaxial accelerometer with a range of ± 200 g (ADXL377, Adafruit Industries, New York City, New York, United States) recorded the acceleration of the forearm at approximately 500 Hz. We placed the device in a wristband on the distal part of the forearm and secured it further using an elastic band. The device recorded accelerations continuously throughout the entire session. After the session, we manually segmented the entire recording into parts containing only a single throw. We performed the segmentation by plotting the entire acceleration signal and locating the 25 peaks each corresponding to a throw. Based on pilot studies, the peak total acceleration appeared to be the best predictor of throwing speed. As such, for each accelerometer segment, we calculated the peak total acceleration by taking the square root of the sum of the acceleration of each axis squared, i.e., finding the Euclidean norm. We obtained the throwing speed in a similar manner to previous handball studies [13,14]. In short: First, we low-pass filtered the marker positions using a two-way second order Butterworth filter with a cut-off frequency of 20 Hz. Second, we calculated the speed of the ball marker using a first-order forward difference. Third and last, we identified the speed of the ball marker at the time of release, which we defined as the moment at which the distance between the ball marker and hand marker increased by more than 1 cm per frame.

### 2.3. Statistical Analysis

We developed four predictive models using multivariable linear regression. One model (Base) contained only the logarithm of the peak total acceleration as predictor, two two-variable models further included throwing type (Type) and sex (Sex), respectively, and one model (Full) contained all three predictors. We used the logarithm of the peak total acceleration rather than peak total acceleration as the predictor as we observed a non-normal distribution of the residuals when using the peak total acceleration.

We used 10−fold cross validation to estimate the internal validity of the predicted throwing speeds [15]. Ten-fold cross validation entails splitting the data into 10 parts (folds), fitting a model to all folds except one and subsequently finding the difference between the observed and predicted speeds (i.e., the prediction error) of the fold that was left out. This procedure is then repeated until all folds have been left out once [16].

We assessed the overall predictive precision of the models by determining the R^2^ statistic and mean absolute error in each hold-out fold. We assessed the ability of the models to produce unbiased prediction by assessing three levels of calibration as described by Van Calster et al. [17]: 1) Mean calibration (calibration-in-the large), which is the difference between the mean predicted and the mean observed throwing speed. 2) Weak calibration (calibration slope), which is the tendency of the model to over- or underestimate the throwing speed, indicated by a calibration slope higher or lower than 1, respectively. 3) Moderate calibration (does the estimated throwing speed correspond to the observed throwing speed), assessed visually by plotting the estimated vs. the observed throwing speeds. With good moderate calibration, points should be scattered around the identity line. Finally, we used linear regression to estimate model coefficients for the entire dataset.

Data and analysis code are available at https://doi.org/10.17605/OSF.IO/YARFP.

## 3. Results

We recorded 475 throws. Speeds of each type of throw are summarized in Table 1.

The Full model was the most precise model, followed by the Type model, the Sex model and finally the Base model (see Table 2). All models were well-calibrated, but the Type and Full models appeared to show slightly better moderate calibration than the Base and Sex models (see Table 2 and Figure 1). Models coefficients for all four models are summarized in Table 3.

## 4. Discussion

The aim of this study was to develop a feasible and cheap method of estimating throwing speeds in handball. To this end, we developed four predictive models that all produced well calibrated and precise estimates. Including sex and type of throw–both individually and in combination–appeared to increase the precision of the prediction models. As such, the proposed models show promise for providing a valid method for estimating throwing speeds in handball.

Compared to previous studies, the throwing speeds of the present study appear similar. For instance, we measured throwing speeds of the standing throw with run-up ranging from 16.8 to 28.1 m/s, while Wagner et al. [18] reported mean throwing speeds of 17.8 and 24.2 for low-level and elite players, respectively. Similarly, Wagner et al. [19] found that the mean throwing speed of the jump throw was 18.0 and 22.3 m/s for low-level and elite players, respectively, while we recorded throwing speeds ranging from 15.4 to 26.8 m/s for the jump throw.

While there were differences in precision between the models, the MAE of all of models were smaller than the measured differences in mean speed between the low, medium and maximal (i.e., jump throws, standing throws with run-up and maximal intensity throws without run-up) throws. Similarly, Plummer et al. [20] found a difference between standing throws with 50% and 100% effort of 5 m/s on average and Wagner et al. [19] showed that the average difference in maximal throwing speed between elite and low-level players was 4.3 m/s. Thus, given the low MAEs, we believe that all of the proposed models have sufficient precision to identify some of the important differences in throwing intensities and player abilities.

Another important aspect of developing a method such as the present is whether the method is usable in practice. We chose to place the device in a regular wristband instead of affixing it more firmly using tape. As wristbands are standard equipment for many players, we believe players are more likely to wear the device in practices and matches than if we had used a custom-made packaging. However, since the wristband was made of elastic weave, the accelerometer could move slightly relative to the forearm, which would introduce errors in the measured accelerations. Such relative movements are most likely to happen when the accelerations are high and thus, we speculate that the prediction error is most likely to be high when the throwing speed is high. A speed-dependent prediction error could be a potential explanation for why we observed a non-normal distribution of the peak total accelerations and subsequently chose to log-transform the accelerations. By fixing the accelerometer more firmly to the forearm, it might be possible to minimize speed-dependent errors and thus provide a more accurate prediction of throwing speed.

We chose the wrist position rather than another position such as on the upper arm for two main reasons: firstly, we speculate that the dorsal side of the wrist is less susceptible to hard tackles, thus limiting the risk for both player and equipment. Secondly, the amount of muscle is minimal at the distal end of the forearm. As such, the circumference changes minimally throughout the game, when compared to a position on the upper arm, where flexing of the muscles can change the circumference to a much larger degree, making it more difficult to fixate the device without being a nuisance to the player.

From a biomechanical perspective, throwing speed is only indirectly related to the risk of shoulder injury as previously outlined. Thus, it might be considering if other measures related to shoulder load, e.g., the raw accelerations at the risk, are a better measure for estimating shoulder injury risk. However, for wearable devices to reach its full potential, users have to adopt and use the device on a regular basis. An indicator of users’ intention to adopt a given technology is the perceived value of a given device. [21] The perceived value can be further broken down into perceived usefulness, perceived enjoyment, and social image which can raise the perceived value and performance and financial risk, which can lower the perceived value. In this regard, we speculate that the perceived usefulness is higher when the device is capable of outputting a measure that users can immediately relate to, namely throwing speed, than if the output is more esoteric to users without a biomechanical background, e.g., acceleration.

When developing prediction models with many predictors, there is always a risk of overfitting. We attempted to minimize the risk by using cross-validation to assess the internal validity, but it should be noted that cross-validation does not formally ensure that findings are externally valid [15]. Further, we deliberately aimed for a diverse set of participants and throwing conditions as previous research has shown that numerous biomechanical parameters in handball throwing depend on gender, level of play and the type of throw [22]. However, to ensure external validity, the results of the present study would have to be replicated in another study considering a different study sample but using the same model coefficients to estimate throwing speed [15].

Another limitation is that the present study only considered a subset of the throwing types most commonly used in handball. Most notably, we did not include the sidearm throw, which is a somewhat common throwing type [14]. Therefore, it is not clear whether the prediction models developed in the present study can validly estimate throwing speed of sidearm throws as well as other, less common throwing variations.

When used in practices and matches, the accelerometer recordings would consist of long streams of data containing accelerations of many throws as well as of many other types of movement. Therefore, it is necessary to divide the stream of data into small segments containing only the recording of a single throw before estimating the throwing speed. In the present study, we manually segmented the entire stream of data, which was a relatively easy task given that each data stream only contained 25 throws clearly separated in time. However, when the data stream contains an unknown number of throws and when there is no clear separation between throws, manual segmentation becomes both time-consuming and potentially unreliable. Therefore, it is important to develop a method that can automatically segment a data stream into parts containing only a single throw before attempting to use the proposed method in settings where the boundaries between throws are not as clear. Furthermore, the models appeared to perform better when including throwing type. Therefore, another future direction for research is to investigate whether throwing type can be automatically discerned from the accelerometer recordings.

The practical appeal of the presented method is substantial for both practitioners and researchers. For instance, monitoring the throwing speed might be a valuable asset for practitioners when employing a progressive return-to-throwing rehabilitation program following shoulder injury, as guidance based on throwing speed appears to provide a better control of joint moment than perceived effort. [11] Likewise, injury researchers obtain a feasible way of estimating throwing speeds during training and match, which can help elucidate the causal underpinnings of shoulder injuries.

In conclusion, we developed four prediction models that can be used to estimate throwing speeds in handball based on a small, wrist-worn accelerometer. The models were well calibrated and precise, implying that the estimates were unbiased and errors were small. Thus, the present method might be a valuable future tool for both practitioners and researchers. However, future research is needed in order to also develop a method for obtaining the number of throws from the same device.

## Figures and Tables

**Figure 1 sensors-20-04925-f001:**
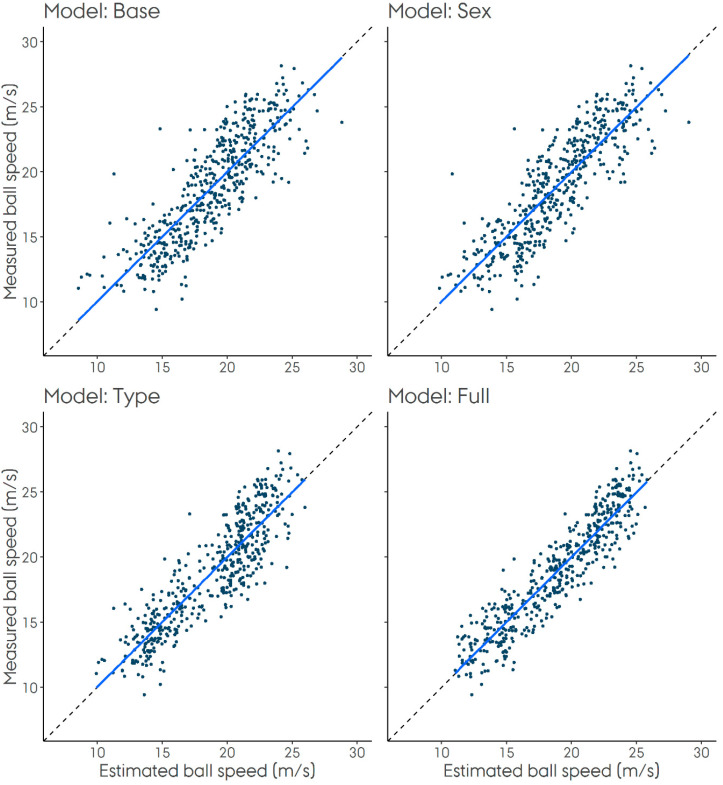
Calibration plots showing estimated vs. observed speeds. The solid, blue line represents the calibration line and the dashed line (only visible at the ends of the solid line) represents a 45° line.

**Table 1 sensors-20-04925-t001:** Summary of throwing speeds for the five different types of throws.

	Low Intensity without Run-Up	Medium Intensity without Run-Up	Maximal Intensity without Run-Up	Maximal Intensity with Run-Up	**Jump Shot**
Mean Throwing Speed (m/s)	13.7	15.8	21.0	22.6	20.9
Range of Velocities (m/s)	[9.4; 17.5]	[11.8; 20.8]	[15.1; 25.9]	[16.8; 28.1]	[15.4; 26.8]

**Table 2 sensors-20-04925-t002:** Performance measures for all models. SD = standard deviation, MAE = mean absolute error.

Measure	Model			
	Base	Sex	Type	Full
R^2^ (SD)	0.71 (0.06)	0.74 (0.06)	0.78 (0.04)	0.86 (0.01)
MAE (SD)	1.82 (0.20)	1.70 (0.16)	1.59 (0.13)	1.30 (0.10)
Weak Calibration	1.00	1.00	1.00	1.00
Mean Calibration	−0.0012	−0.0013	−0.0032	0.0050

**Table 3 sensors-20-04925-t003:** Estimated coefficient values for all models fitted to the entire dataset. CI = confidence interval.

Variable	Model			
	Base	Sex	Type	Full
	β Coefficient (95%−CI)	β Coefficient (95%−CI)	β Coefficient (95%−CI)	β Coefficient (95%−CI)
Intercept	−4.9 (−6.2 to −3.5)	−4.5 (−5.9 to −3.2)	5.6 (3.6 to 7.6)	9.0 (7.4 to 10.7)
Log (Acceleration)	6.4 (6.0 to 6.8)	6.1 (5.7 to 6.5)	4.0 (3.5 to 4.4)	2.7 (2.3 to 3.1)
Sex (male)	--	1.5 (1.1 to 1.9)	--	2.4 (2.1 to 2.7)
Type (Reference Type: Jump Throw)				
Low Intensity without Run-up	--	--	−4.0 (−4.6 to −3.3)	−5.0 (−5.6 to −4.4)
Medium Intensity without Run-up	--	--	−3.0 (−3.6 to −2.4)	−3.7 (−4.2 to −3.2)
Maximal Intensity without Run-up	--	--	−0.5 (−1.0 to 0.1)	−0.3 (−0.8 to 0.2)
Standing with Run-up	--	--	0.6 (0.0 to 1.1)	0.9 (0.5 to 1.4)

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
