# Peer review of "Estimating Throwing Speed in Handball Using a Wearable Device"

_sensors, 2020, doi:10.3390/s20174925_

Round 1

Reviewer 1 Report

I submitted my comments as a pdf file of the submission including yellow markings and comments.

Author Response

We would like to thank both reviewers for their detailed and insightful comments. Both reviewers raise a few major concerns, which we have attempted to address to the best of our abilities, and we believe the manuscript is now substantially improved.

Apart from the specific changes listed below, we have substantially altered the introduction as well as the structure of the discussion. However, due to the size of the changes, we have not included all changes in this cover letter explicitely.

Response to reviewer #1:

L1: We thank the reviewer for the kind words and the suggestion to extend the manuscript to be an article. However, as that would mean that we would have to extend it by almost 100% to hit the journal threshold for article length, we feel that we would have to add superfluous content, which would significantly decrease the manuscript’s quality. Thus, we have chosen not to change the manuscript type.

L12-14: Revised the abstract so it now reads: “Throwing speed is likely a key determinant of shoulder-specific load. However, it is difficult to estimate the speed of throws in handball in field-based settings with many players due to limitations in the current technology. Therefore, the purpose of this study was to develop a novel method to estimate throwing speed in handball using a low-cost accelerometer-based device. Nineteen experienced handball players each performed 25 throws of varying types while we measured the acceleration of the wrist using the accelerometer and the throwing speed using 3D motion capture. Using cross-validation, we developed four prediction models using combinations of the logarithm of the peak total acceleration, sex and throwing type as the predictor and the throwing speed as the outcome. We found that all models were well-calibrated (mean calibration of all models: 0.0 m/s, calibration slope of all models: 1.00) and precise (R2 = 0.71-0.86, mean absolute error = 1.30-1.82 m/s). We conclude that the developed method provides practitioners and researchers with a feasible and cheap method to estimate throwing speed in handball from segments of wrist acceleration signals containing only a single throw.”

L24: Fixed typo: removed final “s” from “throwing speeds”.

L41: Rewrote this section so it now reads: “Shoulder-specific load as opposed to training load in general, is a term used here to describe the overall loads of the structures of the shoulder [4]. It seems biomechanically plausible that the throwing speed could be a key determinant of the shoulder-specific load: higher throwing speeds require larger internal rotation moments at the shoulder [6], larger internal rotation moments at the shoulder require larger muscle forces and ultimately, larger muscle forces cause larger amounts of micro-damage at the tissue level [7]. To our knowledge, there are no studies specifically investigating the relation between throwing speed and shoulder injury risk, but two baseball studies found an association between pitching speed and elbow injury [8], [9]. As such, there are indications that monitoring the speed of throws is important in order to understand and prevent shoulder injuries in handball.”

L51: Added the following: “Nonetheless, wearable devices have been used to estimate ball speed in other sports [10] and given the low cost of wearable devices, such as accelerometers, it would be interesting to investigate the possibility of using accelerometers to determine speed of throws in handball as well”

L57: We agree that automated detection of throws would be valuable and absolutely necessary for implementation in the field (as detailed in L176-188 of the original manuscript). However, we have chosen not to pursue this goal with the current manuscript for two reasons: 1) the current dataset will not allow such a method to be validly developed. The dataset only contains throws and no other handball-related movements. As such, it is impossible to investigate the sensitivity of a potential method with reasonable external validity. 2) We consider the automated detection and speed estimation as somewhat orthogonal projects. For instance, consider a situation where a player turns on the device, makes a single throw and turns off the device again. This could be the case as a part of throwing rehabilitation programs. In such a situation, there is no need for automated detection as the recording only contains a single throw wherefore the present method can be used directly. Likewise, there might be situations where only the number of throws is important, thus rendering the present findings irrelevant. Therefore, we prefer to focus on a single aspect at a time.

To further underline this point, we have modified the abstract so it now reads as detailed in our L12-14 response. Similarly, we have modified the introduction to emphasize the importance of throwing speed, to avoid inadvertently misleading readers into thinking that the automated detection will be dealt with in this article.

L59: The Instructions to Authors explicitly state with respect to the Introduction that “Finally, briefly mention the main aim of the work and highlight the main conclusions”. As such, we have added the main conclusion to the end of the introduction.

L77: Clarified the role of reflective markers, so the sentence now reads: “During each throw, we recorded the position of a reflective marker on both the hand and the ball using 3D motion capture (8 cameras recording at 240 Hz; ProReflex MCU1000, Qualisys AB, Gothenburg, Sweden)”

L79: We thank the reviewer for the suggestion to also compare 3D measured acceleration of the hand with the acceleration of the wrist/forearm recorded by the device. However, due to the different anatomical locations (hand vs wrist/forearm), we find that these two accelerations are unlikely to be comparable. Therefore, we have abstained from performing such a comparison.

L82: Since recording the data for the present study, we have further developed the device/packaging and are unfortunately no longer in possession of the old version. Therefore, unfortunately we cannot produce a figure, which accurately depicts the setup described in the manuscript.

L85: See comments to L57.

L86-87: We have added the following sentence to explain why we chose the peak total acceleration as predictor: “Based on pilot studies, the peak total acceleration appeared to be the best predictor of throwing speed.”

L96: We have added the following sentence to explain why we used the logarithm of the peak total acceleration: “We used the logarithm of the peak total acceleration rather than peak total acceleration as the predictor as we observed a non-normal distribution of the residuals when using the peak total acceleration.”

L97: We agree that the reliance on throwing type is noteworthy, hence the recommendation in L186 that future research should investigate methods to determine this. However, as with the automated detection, we believe that this is outside the scope of the present article.

L100: We are somewhat uncertain if the reviewer asks for the rationale for using cross validation or for using ten folds. We have added a reference to D. G. Altman, Y. Vergouwe, P. Royston, and K. G. M. Moons, “Prognosis and prognostic research: validating a prognostic model,” BMJ, vol. 338, no. may28 1, pp. b605–b605, May 2009 to support the rationale for using cross validation to assess internal validity: “We used 10-fold cross validation to estimate the internal validity of the predicted throwing speeds [11].”

L104: We thank the reviewer for the suggestion, but respectfully disagree that ICC is a better performance measure than R2 in the present study. Since we are interested in the predictive performance of our models, we compute R2 rather than ICC, since ICC – as we understand it – is more related to the reliability of measurements.

L121: The table of throwing speeds is included as the difference in throwing speeds are used in the discussion, thus we prefer to retain this table. Furthermore, it is indeed confusing that a speed of 23.4 m/s classifies as a short pass. After the reviewer’s comment, we rechecked the data and the in-house developed analysis code, and discovered an unfortunate (and embarrassing) coding error, where the type of a short pass and a standing throw with run-up was swapped. Consequently, we have re-run the analysis, which have slightly improved R2 and MAE of the Type and Full models. We have also updated the calibrations plots, Table 1, 2 and 3 and the corresponding numbers in the discussion. However, the fundamental results and conclusions has not changed. We thank the reviewer for discovering the inconsistency, which led us to discover the coding error.

L130: Since the type is a categorical variable, the coefficients for the throwing types are in relation to a reference type, which in this case is the jump throw. To clarify, we have changed “ref” to “reference type”.

L133: Reworded sentence to: “To this end, we developed four predictive models that all produced well calibrated and precise estimates”

L139: Reworded sentence to: “While there were differences in precision between the models, the MAE of all of models were smaller than the measured differences in mean speed between the low, medium and maximal (i.e. jump throws, standing throws with run-up and maximal intensity throws without run-up) throws”

L148: Added the following sentence: “However, to ensure external validity, the results of the present study would have to be replicated in another study considering a different study sample but using the same model coefficients to estimate throwing speed [11].”

L150: Added text to emphasize that the comment was related to handball throwing: “that numerous biomechanical parameters in handball throwing depend on gender, level of play and the type of throw [19].”

L152: Moved paragraph up as suggested by reviewer and reworded first sentence to: “Compared to previous studies, the throwing speeds of the present study appear similar.”

L162: Removed line break.

L167: Corrected as mentioned in the response to L96.

L173: Added the following text: “[…], as guidance based on throwing speed appears to provide a better control of joint moment than subjective measures such as perceived effort. [19]”

L189: Fixed typo.

Reviewer 2 Report

Please see the attached file for comments

Author Response

We would like to thank both reviewers for their detailed and insightful comments. Both reviewers raise a few major concerns, which we have attempted to address to the best of our abilities, and we believe the manuscript is now substantially improved.

Apart from the specific changes listed below, we have substantially altered the introduction as well as the structure of the discussion. However, due to the size of the changes, we have not included all changes in this cover letter explicitely.

Response to reviewer #2:

Title: We agree with the reviewer’s comments that “A Novel Method” does not add anything substantial to the title. Consequently, we have changed the title to “Estimating Throwing Speed in Handball using a Wearable Device” as per the reviewer’s suggestion.

General comments: The reviewer raises an interesting point which has also been a ground for some discussion amongst the authors previously: what does throwing speed add that acceleration alone cannot tell us? From a biomechanical perspective, the answer might very well be: nothing (given that shoulder loads, i.e. forces, are directly related to acceleration, not velocity). However, from a practical perspective, we believe there are significant upsides to speed rather than acceleration. Firstly, the vast majority of the current literature on rehabilitation and injury associations (just not for shoulder injuries) in throwing sports uses throwing speed (see added paragraph in introduction). Secondly, physical therapists, players and coaches are more likely to use a certain device if they get an output that they are familiar with and perceive as being useful (see added paragraph in discussion). In this regard, we believe speed is better than acceleration as we speculate that practitioners without a biomechanical background will have a difficult time interpreting and acting on acceleration values. Nonetheless, this is an interesting discussion and we hope that our revision provides a sufficient justification of measuring throwing speed in regards to injury research and rehabilitation.

L40: We have removed this line in the updated manuscript.

L41-44: We have modified the introduction to more clearly state the reasoning behind the link between throwing speed and shoulder-specific load. Furthermore, we have added a paragraph about the importance of measuring throwing speed when rehabilitating shoulder injuries.

Segment lengths: Due to the nature of the trial protocol, the segment lengths varied considerably as the only requirement was that a segment only contained a single throw. As such, the segment lengths varies from ~4 seconds to ~100 seconds. We agree with the reviewer that this information is not necessary for the manuscript and we have therefore left it out.

Results: Updated Figure 1 to match reviewer’s suggestions.

L158: We have added the following paragraph to the discussion: “We chose the wrist position rather than another position such as on the upper arm for two main reasons: firstly, we speculate that the dorsal side of the wrist is less susceptible to hard tackles, thus limiting the risk for both player and equipment. Secondly, the amount of muscle is minimal at the distal end of the forearm. As such, the circumference changes minimally throughout the game, when compared to a position on the upper arm, where flexing of the muscles can change the circumference to a much larger degree, making it more difficult to fixate the device without being a nuisance to the player.” It is also worth noting that the IHF regulations require the device to be wrapped in protective padding no matter where the device is placed: https://www.ihf.info/sites/default/files/2019-07/0_Regulations%20on%20Protective%20Equipment%20and%20Accessories_GB.pdf

Throwing types: We have added the following paragraph to the discussion: “Another limitation is that the present study only considered a subset of the throwing types most commonly used in handball. Most notably, we did not include the sidearm throw, which is a somewhat common throwing type [12]. Therefore, it is not clear whether the prediction models developed in the present study can validly estimate throwing speed of sidearm throws as well as other, less common throwing variations.”

Dataset: The reviewer is indeed correct that PeakX, PeakY and PeakZ are not used in the analysis. Therefore, we have removed these columns from the dataset and updated the file repository. We would also like to applaud the reviewer for being so thorough and spotting the error.

Round 2

Reviewer 1 Report

Though i still detect some flaws that have not been adressed adequately i can accept the paper as presented.

I would have been happier with a "real" major revision and an extented content. 

Reviewer 2 Report

Thank you for you answers to my questions. 

I am still not convinced about the reasoning around throwing speed and shoulder load. Especially the sentence "Throwing speed is likely a key determinant of shoulder-specific load" in the abstract is to harsh IMO. 

Although I dont agree on the reasoning behind the manuscript, I think the manuscript is good. After all, you do argue for your view, and even though I`m not convinced - I could just as well be wrong. I therefor suggest the editor to accept the manuscript.